# Structural annotation of unknown molecules in a miniaturized mass spectrometer based on a transformer enabled fragment tree method
Yiming Yang [1], Shuang Sun[1], Shuyuan Yang[1], Qin Yang[1], Xinqiong Lu[2], Xiaohao Wang[1], Quan Yu [1], Xinming Huo [3] ✉ & Xiang Qian [1] ✉

Structural annotation of small molecules in tandem mass spectrometry has always been a central challenge in mass spectrometry analysis, especially using a miniaturized mass spectrometer for on-site testing. Here, we propose the Transformer enabled Fragment Tree (TeFT) method, which combines various types of fragmentation tree models and a deep learning Transformer module. It is aimed to generate the specific structure of molecules de novo solely from mass spectrometry spectra. The evaluation results on different open-source databases indicated that the proposed model achieved remarkable results in that the majority of molecular structures of compounds in the test can be successfully recognized. Also, the TeFT has been validated on a miniaturized mass spectrometer with low-resolution spectra for 16 flavonoid alcohols, achieving complete structure prediction for 8 substances. Finally, TeFT confirmed the structure of the compound contained in a Chinese medicine substance called the Anweiyang capsule. These results indicate that the TeFT method is suitable for annotating fragmentation peaks with clear fragmentation rules, particularly when applied to on-site mass spectrometry with lower mass resolution.

The analysis of small molecular compounds' structure in mass spectrometry and predicting the molecule's structure to be measured from tandem mass spectrometry spectra are primary research targets in analytical chemistry. This objective is especially relevant to discovering new homologous derivatives, natural product research, non-targeted metabolomics, drug research, food safety, pharmaceutical ingredient analysis, and drug detection[1–5]. Especially in on-site testing, the timely identification of new drugs and psychoactive substances via miniaturized mass spectrometers is increasingly needed. Usually, new psychoactive substances have similar chemical structures to existing drugs, and a series of derivatives are produced through some chemical modifications[6–8]. The rapid use of on-site mass spectrometers for the detection of such substances' structures is crucial in preventing them from entering widespread circulation in the market without permission. Meanwhile, within the field of traditional Chinese medicine[9–12], the rapid comprehension of the chemical composition and action mechanisms of herbal medicines through on-site mass spectrometers

can contribute significantly to expanding the acceptance and utilization of these medicines by a broader population. Due to the chemical diversity of these compounds' structures and the limited mass resolution of miniaturized mass spectrometers, on-site determination of the structures of unfamiliar substances faces significant challenges. It is necessary to establish a model using low-resolution mass spectrometry spectra to predict the molecule's structure.

One common approach to automatically interpret MS$^n$ spectra is to search in a mass spectrometry database[13–16]. This methodology involves comparing the mass spectra of compounds under specific conditions with a database containing a large number of reference mass spectra. Through an algorithmic calculation of similarity, the molecule corresponding to the most similar spectrum is identified in the database. McLafferty et al. proposed a probability-based matching system that utilizes peak occurrence probability and empirical correction to accurately sort candidate molecule lists[17]. Similarly, Roman Mylonas et al. introduced the X-Rank algorithm to

[1]Shenzhen International Graduate School, Tsinghua University, Shenzhen 518055, China. [2]CHIN Instrument (Hefei) Co., Ltd., Hefei 231200, China. [3]Key Laboratory of Sensing Technology and Biomedical Instruments of Guangdong Province, School of Biomedical Engineering, Shenzhen Campus of Sun Yat-sen University, Shenzhen 518107, China. ✉e-mail: huoxm@mail.sysu.edu.cn; qian.xiang@sz.tsinghua.edu.cn

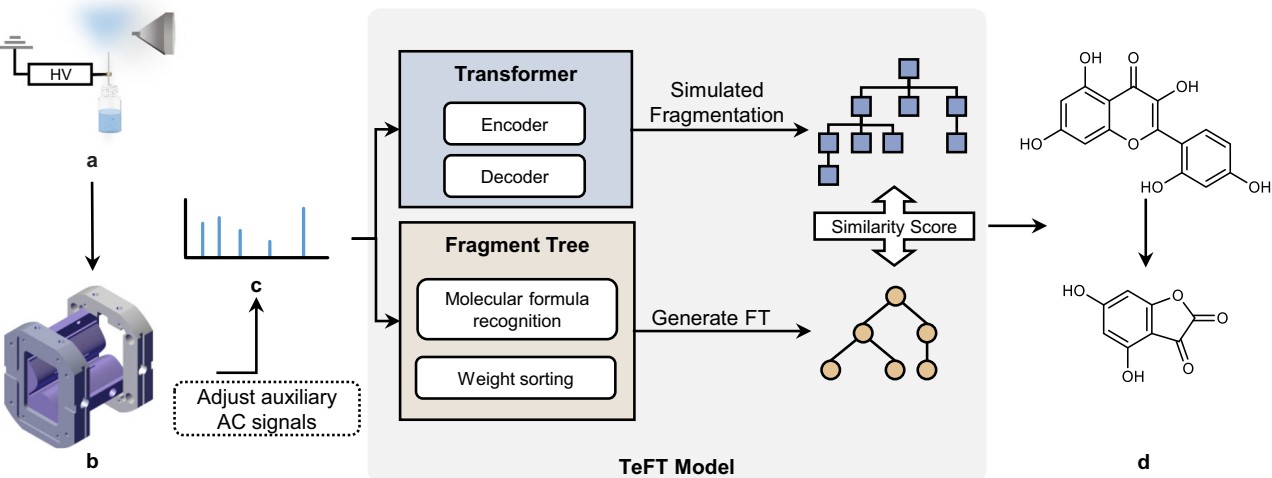

**Fig. 1 | A conceptual overview of TeFT.** In the on-site application, miniaturized mass spectrometry is used to get MS$^n$ spectra. Using the Transformer and Fragment tree generation approach, a range of potential SMILES strings and the fragment tree were predicted. Through the simulation of fragmentation, several SMILES fragmentation trees are generated and then subjected to a comparative analysis and scoring process against the fragmentation tree. The SMILES tree with the highest score provides possible annotations for each peak in the spectrum. **a** Electrospray ionization. **b** Miniaturized ion trap. **c** MS$^n$ spectrum. **d** Candidate annotation of fragmentation peaks.

rank peak intensities of mass spectra, establish correlations between different mass spectra, determine the probability of matching with mass spectra from a reference library, and enable cross-mass spectrometry platform recognition and search[18]. In 2016, Christoph Ruttkies proposed the MetFrag model, combining database search algorithms and fragment prediction algorithms for identifying the structure of small molecules from tandem mass spectrometry data. This method aids in the identification of compounds not yet included in mass spectrometry databases. It involves filtering and scoring candidate structures based on matched peaks' mass-to-charge ratio, intensity, and bond dissociation energy, thereby enhancing the ability to recognize unknown compounds[19,20]. The SIRIUS series methods[21–23], proposed by Sebastian Böcker et al., are considered to be a more effective mass spectrometry database search algorithm. This method combines high-resolution isotope pattern analysis, fragment tree (FT), and CSI: FingerID[21] to assist in searching molecular structure databases. One significant limitation of mass spectrometry library search techniques is their inability to identify unknown natural products and drug metabolites. Meanwhile, in the case of low-resolution spectra acquired from a miniaturized mass spectrometer used for on-site testing, it is difficult to provide corresponding databases for searching.

In addition, machine learning and the deep learning model have been applied to analytical chemistry and drug structure design for a long time[24–30]. One of the great advantages of deep learning models is that they can generate molecular structures from mass spectrometry spectra without being given explicit rules. The deep learning model encodes and decodes chemical substances through methods such as SMILES (simplified molecular-input line-entry system)[31] strings and molecular graph construction[32,33], which can then transform the annotation of mass spectrometry structure into language translation or graph neural network problems. Böcker's research team proposed a model called CANOPUS that combines SVM and deep neural networks[34]. DNN predicted compound categories from fingerprints and completed compound classification. Aditya et al. proposed MassGenie, a Transformer-Based deep learning method[35]. This method transforms the molecular recognition problem into a language translation problem, where the source language is a list of high-resolution mass spectra peaks, and the translation language is the SMILES strings of the molecule. Meanwhile, the DarkNPS model is based on the LSTM model for automatic structural analysis of new psychoactive substances[8]. In 2022, Michael et al. constructed the model MSNovelist using an encoder-decoder neural network to achieve de novo prediction of the structure of unknown compounds from tandem

mass spectrometry[36]. This model combines fingerprint prediction with neural networks for the annotation of molecular structures. The deep learning model is limited by the computing resources of hardware platforms and the limited availability of MS$^2$ spectra, and its testing performance varies across different network architectures. MassGenie utilizes a network with over 400 million nodes, completing training on the DGX A100 8-GPU system. However, considering the on-site applications with a miniaturized Mass Spectrometer platform poses challenges in achieving large-scale model training similar to MassGenie.

This paper proposed a so-called "Transformer enabled Fragment Tree (TeFT)" framework to identify the unknown molecular structures for tandem mass spectrometry; it was composed of a simulated semantic fragment tree model (SMILES tree) generated through the deep learning Transformer module[37] and the FT[38–41] directly generated through the original MS$^n$ spectral data. By aligning and comparing the similarity of the two trees, the molecular structure of the tested chemical substance with the highest possibility can be predicted. This method can be embedded into any tandem mass spectrometry systems with fragmentation function; however, it is particularly suitable for miniaturized mass spectrometry for on-site applications where the spectral resolution is limited. Furthermore, a relatively lightweight Transformer module with 65 million nodes was adopted in the current work, thus, the computational complexity is also suitable for on-site applications.

Figure 1 illustrates the conceptual workflow of the proposed method. All the experiments were applied on a miniaturized ion trap mass spectrometry with a self-aspiration capillary electrospray ionization source (SACESI) that we have previously developed[42–45]. MS$^n$ spectra were obtained using high-resolution isolation and collision-induced dissociation (CID) sequences by carefully controlling the frequency and amplitude of the auxiliary AC signal applied to the ion trap[46]. Experimental details can be found in the method section.

The original MS$^n$ spectral data are sorted according to peak intensities, and several fragments with the highest intensity are selected as inputs for the deep learning Transformer module. The Transformer module consists of several encoder and decoder layers, utilizing a large number of open-source libraries to learn the potential relationship between molecular SMILES strings and tandem mass spectrometry data with the assistance of attention mechanisms. Inputting an MS$^n$ spectral data into the Transformer, the module will output a list of SMILES strings for the molecule, corresponding to the possible chemical structure of the unknown substance. It's worth

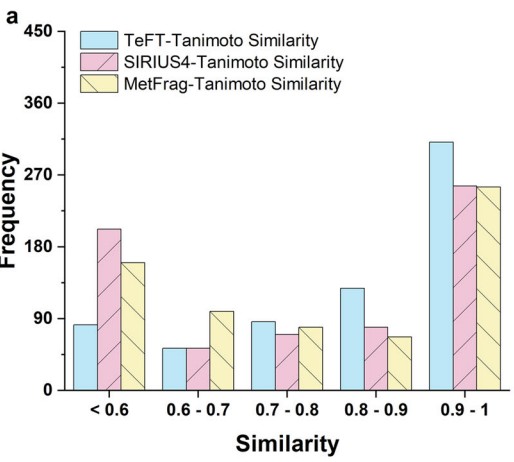
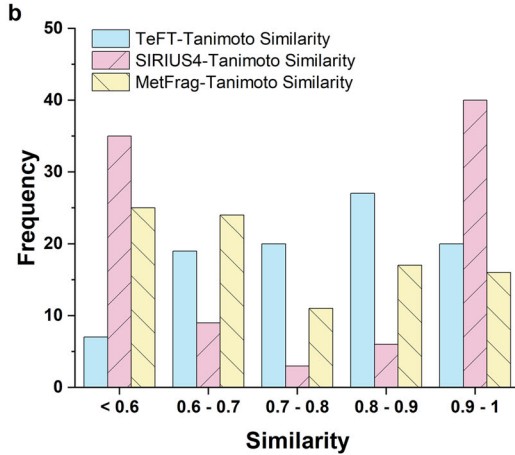

**Fig. 2 | The Tanimoto similarity of the best incorrect candidate to correct structure for TeFT, SIRIUS4, and MetFrag in different datasets. a** Test set. **b** CASMI 2017 challenge.

noting that due to the low resolution of the original spectra, the reduction in model parameters, and the adoption of a more lightweight model architecture, the outputs of the model are not unique across multiple runs, necessitating the sorting of results. Next, the candidate substances in the list undergo the simulated fragmentation to generate a series of SMILES trees. During the process of simulated fragmentation, we adopted the general fragmentation rules that often occur in chemical bonds in mass spectrometers (as listed in Supplementary Table S1); we also provide an interface for adding specific new rules into the fragmentation process by SMART (SMILES Arbitrary Target Specification). The resulting SMILES trees are composed of tree nodes with the SMILES strings of molecules or fragments and the loss of each fragmentation formula as edges. The SMILES tree represents the most possible dissociation scenarios of the specific molecule in tandem mass spectrometry.

Also, the traditional FT algorithm generates the corresponding FTs from the original $MS^n$ spectral data directly. After guessing the formula of each fragment, we calculate a reasonable loss for each fragmentation peak in the $MS^n$ spectrum and add corresponding weights. By comparing each node and corresponding losses between the SMILES tree and FT, we can score the similarity of the two trees. Our results indicated that the SMILES tree can determine the most comparable outcomes to the tested substance, including the structural annotations of individual molecules. It can be postulated that the SMILES tree with the highest evaluation score is highly probable to be the tested substance. Meanwhile, the possible dissociation pathways of the substance contained in the SMILES tree with the highest score also provide possible annotations for each fragmentation peak in the spectrum.

Compared to library searching methods such as the SIRIUS4 and MetFrag, the proposed model achieved better results in that the majority of molecular structures of compounds in the test can be successfully recognized. Additionally, 23 molecule experiments were conducted via miniaturized linear ion trap mass spectrometers, and the complete structures of ten flavonoids and two stilbenes were successfully predicted by the model. Finally, to demonstrate TeFT on real-drug data, the model identified one of the main components of traditional Chinese medicine Anweiyang capsule.

## Results

### Model validation

We tested the Transformer model's accuracy by utilizing two separate databases. We randomly extracted 660 non-repetitive $MS^2$ spectra from the mass spectrometry library and used them as the test set. For every $MS^2$ spectrum, we ran the model 100 times to predict various candidate substances. After eliminating invalid SMILES strings, we then assessed the similarity between the predicted results and the molecular fingerprints of actual structures, using the Tanimoto similarity methods in RDKit[47]. Molecular fingerprinting is a technique for representing molecules as

mathematical constructs. This method enables the mapping of molecules into a vector space by considering their distinctive features, including functional groups, atomic sequences, and various topological structures. Molecular fingerprinting is widely employed to facilitate similarity comparisons among molecules[48]. In this study, we employed MACCS and Morgan fingerprints to create molecular fingerprints. The performance of the two datasets is depicted in Fig. 2, illustrating the distribution of molecular fingerprint similarity between the model's best candidate output and the actual substance. The predicted molecular fingerprint similarity data for all substances is available in Supplementary Data 1. In the test set, the Tanimoto similarity method revealed that the Transformer model correctly identified 30% (195/660) of the actual structures, all with a Tanimoto similarity value of 1. The percentage of Tanimoto similarity greater than 0.9 was 47% (311/660), while the percentage greater than 0.8 was 67% (439/660). In comparison, we used SIRIUS4 and MetFrag to search 660 mass spectra and select the substance with the highest ranking as the prediction result. The results showed that SIRIUS4 correctly predicted the entire structure of 27.6% of the substance, while 39% had fingerprint similarity greater than 0.9. The predicted results of the MetFrag model are relatively close to those of SIRIUS4. This suggests that the Transformer model efficiently employs the spectrum's structural data.

The second benchmark comprised 93 positive-mode $MS^2$ spectra from the CASMI 2017 contest (http://www.casmi-contest.org/2017/index.shtml), a typical method for evaluating model performance. The same process as in the previous test set was used to evaluate the $MS^2$ spectra. Of the 93 substances, 12% predicted all structures correctly and 22% accurately predicted most structures. Notably, different molecular fingerprint similarity computation methods (Dice similarity) can increase the percentage of structures with a fingerprint similarity greater than 0.8 to 78%, and a similarity greater than 0.9 to 39%. Similarly, SIRIUS4 prediction results show that 43% accurately predicted most structures and The MetFrag model predicted the majority of structures for 17.2% of the substances.

By comparing with the SIRIUS4 and MetFrag, we can find that the Transformer model can generate de novo chemical structures solely from tandem mass spectrometry and generate candidate lists of substances that are more similar to authentic compounds. The Wilcoxon signed-rank test was employed for non-parametric statistical analysis, and the final $p$ values are reported in Table 1. These results indicate a significant difference in the testing performance of the TeFT model compared to the SIRIUS4 model and the MetFrag. Additionally, in Supplementary Fig. S4, we provided the top-5 accuracy results for two benchmark methods and the TeFT model. The test results indicate that, compared to the other two models, the TeFT model demonstrates superior predictive capabilities on the test set. The top-k rankings also show that the TeFT model's predicted results consistently rank at the forefront, affirming the rationality of the model's ranking

algorithm. On the CASMI 2017 dataset, the predictive ability of the TeFT model is slightly below SIRIUS4, comparable to MetFrag, but the overall predictive accuracy of TeFT is much higher than the other two methods. The majority of results' similarities predicted by TeFT are above 0.8. Therefore, the model exhibits robust predictive capabilities across different datasets. For some mass spectrometry databases that lack annotations, the TeFT model can provide references for possible molecular structures.

To validate the effectiveness of the fragment tree scoring mechanism, we use the SIRIUS4 and MetFrag methods as replacements for the Transformer component of TeFT. We evaluated their similarity with the predictions of SIRIUS4 and MetFrag. The results are provided in the Supplementary Table S6. Among the substances ranked first in the original SIRIUS4 scoring, 56% were also ranked first in the fragment tree scoring, while MetFrag had a proportion of 16%. Additionally, SIRIUS4 retrieved

27.9% of substances in the fragment tree scoring list, while MetFrag had 26.7% (ranked top 1). The Transformer model demonstrated higher flexibility in generating substances and produced more similar molecules. Fragment tree similarity scores also indicated a certain level of reliability. It is essential to note that only the TeFT model is feasible for low-quality resolution spectra.

## Model performance in miniaturized linear ion trap mass spectrometer

To evaluate the model's predictive ability for various drug types, we purchased 23 substances, including flavanols, stilbenes, flavones, and Rotundine. Fig. 3 provides an illustration of the prediction process of the entire model using Galangin as an example. All information regarding the tested substances and the detailed test results are presented in Supplementary Data 2. The Transformer model generated a series of candidate substances. By simulating their decomposition into SMILES trees and generating FT, the model scored the similarity between these trees and identified the closest substance structure from the results and it provides potential structural annotations for the fragmentation peak. The experimental results are presented in Table 2. Out of the 16 flavanol drugs, eight achieved precise structural prediction, with a molecular fingerprint similarity of 1 (three ranked first). Thus, by employing the FT similarity algorithm, we can reliably predict and determine all the genuine structures of the substance,

**Table 1 | The *p* values resulting from the Wilcoxon signed-rank non-parametric statistical test on the similarity results between TeFT and SIRIUS4, MetFrag**

| P values | | SIRIUS4 | MetFrag |
|---|---|---|---|
| TeFT | Test | $3.8 \times 10^{-13}$ | $8.2 \times 10^{-9}$ |
| | CASMI 2017 | $4.4 \times 10^{-6}$ | $2.7 \times 10^{-11}$ |

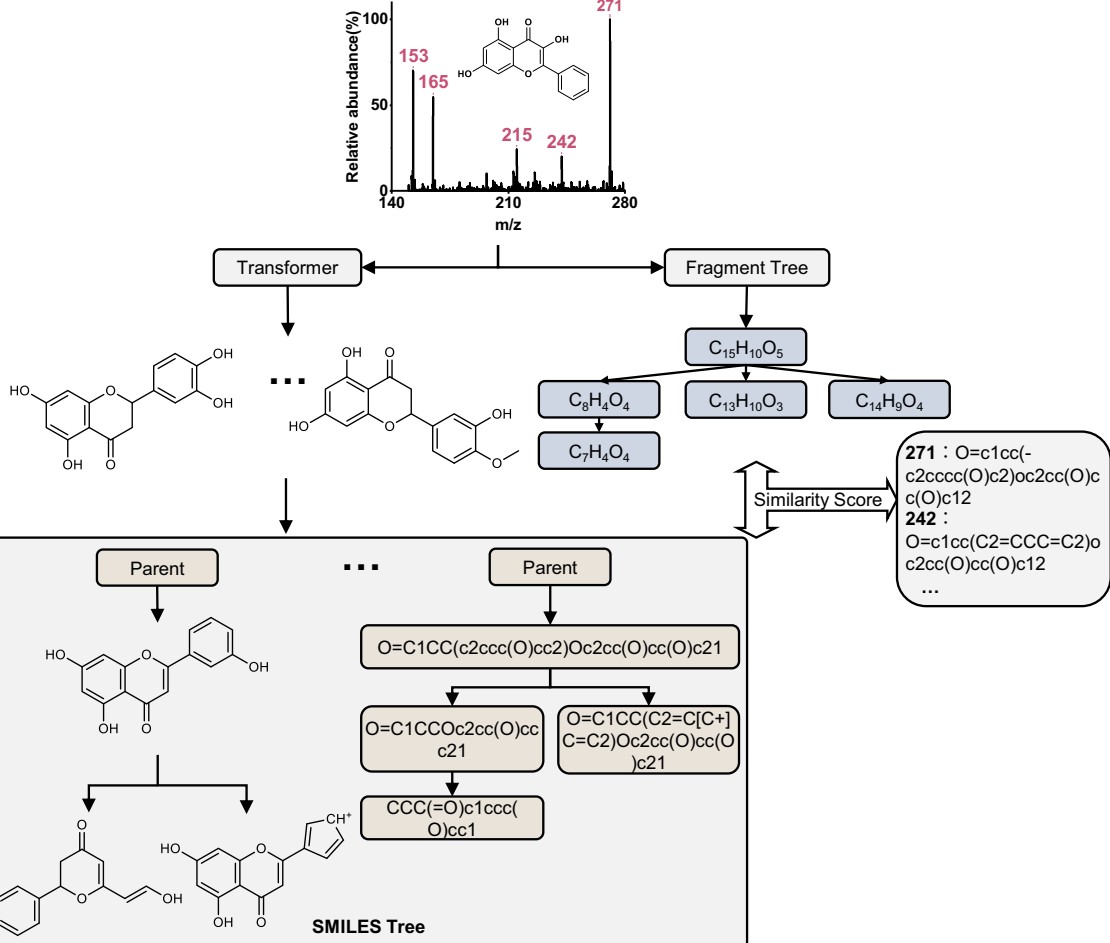

**Fig. 3 | Mass spectrometry data prediction process (taking Galangin as an example).** Four fragment peaks in the MS$^2$ spectra along with the parent ion peaks m/z = 271 of Galangin, were fed into two distinct models: the Transformer model and the fragmentation tree generation model. The Transformer model, in its predictive role, generates a list of potential molecules. For each molecule in this list, it then conducts simulating fragmentation based on the potential cleavage patterns of

flavonoids in mass spectrometry, resulting in the creation of a SMILES tree. Subsequently, a similarity score is assigned by comparing the SMILES tree to the fragmentation tree. The highest score achieved designates the most probable substance, along with its potential SMILES expression corresponding to the fragmentation peak.

## Table 2 | Test results for four types of substances

|  | Amount | Minimum similarity | Quantity of completely similar substances |
|---|---|---|---|
| Flavanols | 16 | 0.96 | 8 |
| Stilbenes | 4 | 0.93 | 2 |
| Flavones | 2 | 0.98 | 1 |
| Others | 1 | 1 | 1 |

The minimum similarity refers to the molecular fingerprint similarity between the worst-predicted result and the real molecule among all predicted results in that category. The number of completely similar substances refers to the quantity of substances in that category for which the predicted result is identical to the real molecule.

alongside possible structural annotations of each fragment, with the correct answer being ranked first among candidate substances. The remaining substances, including flavanones and stilbenes, have similarly obtained an accurate prediction. The structure of one flavanone and two stilbenes has been completely identified. It is worth mentioning that concerning the other substances tested, their molecular fingerprint similarity is mainly distributed around 0.97 (0.93 being the worst outcome). The distribution of ranking scores based on the fragment tree scoring mechanism of TeFT for four classes of drugs is illustrated in Fig. 4. Figure 4 indicates that this scoring approach guaranteed that the most similar substances consistently occupy the top three positions among the candidate substances.

The experimental study indicated that, based on the predicted outcome of the Transformer model, the top three substances that were most similar to the tested substance were identified by SMILES tree generation and subsequent similarity scoring with the FT. This methodology enables the selection of the most similar candidate substance to the original mass spectrum, while also providing a potential structural interpretation for the fragmentation spectrum. Specifically, Simulated fragmentation can break up the redundant parts of the measured molecular results to make them closer to the original molecule. Moreover, this approach provides an alternative means of comparing substance similarity, eschewing the use of molecular fingerprints and possessing a certain level of credibility.

Furthermore, the experimental results demonstrate the applicability of the TeFT model on a low-resolution mass spectrometry platform. Despite the low mass resolution of the mass spectrometry, the model's reliability in making structural predictions and small molecule annotations remains unchanged. Accordingly, these findings significantly ease the processing of mass spectrometry spectra acquired by miniaturized mass spectrometers and broaden their application scope in on-site detection.

### Determination of unknown drug ingredients

The TeFT model employed in this study can identify unknown components in drugs. Anweiyang Capsule, widely used for the treatment of gastric and duodenal ulcers, is primarily composed of flavonoids extracted from liquorice. Following several preprocessing steps in the experimental procedure (details provided in the Methods), the flavonoids contained within the capsules can be successfully extracted and subsequently subjected to analysis using a miniaturized ion trap mass spectrometry.

Upon obtaining the full spectrum of the substance, as Fig. 5a shows, the next step involved the isolation and fragmentation of the marker peak with an m/z value of 269. Fig. 5b illustrates that the MS² spectrogram of the substance contains a total of three fragmentation peaks: 213, 237, and 254. This spectrogram was then fed into the TeFT model, which processed it to generate a series of SMILES trees, each accompanied by a similarity score of two trees. The possible structural annotations for the three fragmentation peaks and the parent ion peak are depicted in Fig. 5c. These scored SMILES trees provided valuable insights into the structural characteristics of the substance. The highest-scoring SMILES tree supplied complete annotations for three peaks in the fragmentation spectrum (i.e., the predicted molecular formula of the fragmentation tree is identical to the actual molecular formula), and incomplete annotations for the remaining peak (albeit with the same type and number of elements, except for H). The best-scoring

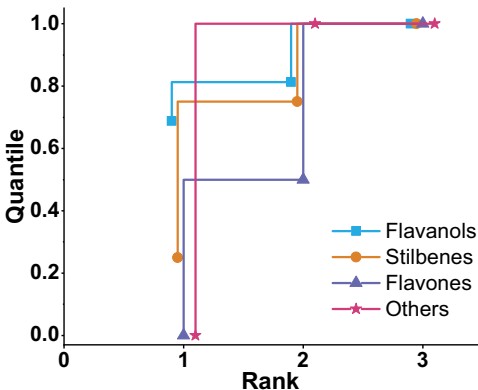

**Fig. 4 | The distribution of ranking scores based on the predictive scoring of MS²
spectra for four classes of substances tested on-site using the TeFT model.** By
employing a fragment tree scoring mechanism, the results were ranked, showcasing
the distribution of ranking positions for the most similar substances. This scoring
method ensures that the most similar substances consistently occupy the top three
positions among the candidate substance.

substance in the candidate list was formononetin. Consequently, we hypothesize that Anweiyang capsules contain flavonoids that comprise formononetin. Relevant literature has proved that formononetin is indeed one of the main components of the Anweiyang Capsule[49]. For other components, the TeFT model can provide a series of predicted molecular structures. However, due to the lack of clear evidence in other literature and existing resources to confirm the substances corresponding to the remaining peaks, the predicted structures for m/z = 262 and 249, as Fig. 5a shows are provided for reference purposes only. We compared the TeFT model predictions based on the miniaturized MS method with the test results from commercial high-resolution MS (Thermo Fisher Q-Extractive Orbitrap) using the same extracted solution. High-resolution mass spectra have been provided in the Supplementary Fig. S5. We observed that the molecular formulas of the top two structures predicted by the model were consistent with the measured molecular formulas from the commercial high-resolution MS results. Our predictive results exhibit molecular formulas that match the measurements obtained from high-resolution mass spectrometry with very high mass accuracy (the errors are less than 0.01). For the peak at m/z = 262, our predictive model calculated a theoretical molecular mass of 262.191 ($C_{15}H_{24}N_3O^+$), which aligns precisely with the high-resolution measurement result of 262.191. The peak at m/z = 249 exhibits a comparable predictive accuracy, with the predicted molecular formula being $C_{16}H_{13}N_2O^+$ and the theoretical molecular weight being 249.102, which aligns closely with the high-resolution measurement result of 249.111. Because even small variations of m/z value can indicate different molecular formulas, high-resolution MS peaks can be used to determine the molecular formulas of the analytes in practical applications. This, to some extent, corroborates the accuracy of our model predictions based on low-resolution MS data. However, fully structural accuracies of the predicted molecules should be further verified using multiple analytical methods in the future.

The experiment illustrated that the model can generate detailed structures of flavonoids found in unknown drugs and provide dependable structural annotations for drug fragmentation peaks, using tandem mass spectrometry data alone.

## Discussion

We propose a novel prediction model, TeFT, designed for de novo structure generation from low-resolution tandem mass spectrometry (MSⁿ) spectra and partial structural annotation of mass spectrometry peaks. TeFT combines the deep learning Transformer model with a modified fragmentation tree generation algorithm, incorporating an extensible fragmentation rule library to ensure versatility for various substances. By simulating molecular

**Fig. 5 | Experimental determination of unknown drug components. a** The detection spectrum of substances in the Anweiyang capsule. **b** Its fragmentation spectrum. **c** The possible structural annotations corresponding to each spectral peak inferred by the TeFT model.

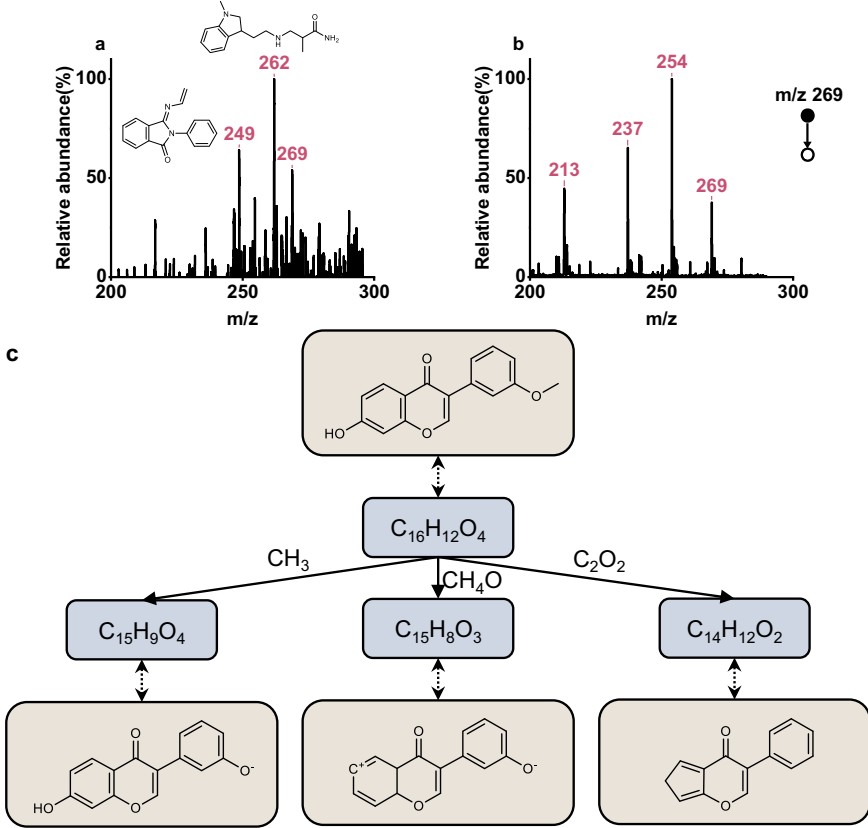

fragmentation based on deep learning predictions and scoring the similarity with fragmentation trees, TeFT can identify the most probable structures from the multiple results predicted by the Transformer model. We found that the similarity scores between the two fragmentation trees can serve as a novel metric for assessing molecular similarity and maintaining a high level of confidence.

Specifically, we adopted a rule-embedded Transformer model. Experimental results from on-site mass spectrometry indicate that due to the limitations of spectral resolution, mass accuracy, and model architecture, the Transformer model does not yield unique output for the same input. During several different inference processes, we may obtain several similar but not entirely identical molecular representations. These molecules have very close molecular weights and highly similar structures. Therefore, further filtering is required to eliminate this "ambiguity." To enable similarity ranking of candidate substances, this study established a similarity scoring mechanism using both molecular fragment tree and SMILES fragment tree models.

Firstly, a molecular fragment tree model was constructed based on tandem mass spectrometry spectra. In the molecular fragment tree, each node corresponds to a molecular formula of a fragment peak in the spectrum, with edges connecting pairs of nodes indicating lost functional groups. The molecular fragment tree generates the maximum weighted fragment tree using a combination of chemical and probabilistic models, representing the most likely fragmentation pathways and outcomes for the parent ion.

Subsequently, based on the molecular fragment tree model, we designed a SMILES fragment tree model using the Recap method. The SMILES fragment tree model simulates the fragmentation of molecules according to possible fragmentation rules in the mass spectrometer, incorporating structural information of substance fragments. The SMILES fragment tree integrates molecule representation and fragmentation rules, with nodes labeled with SMILES strings of fragments and edges indicating potential fragmentation patterns in the mass spectrum, labeled with molecular formulas of the lost structures. The SMILES fragment tree represents possible dissociation scenarios of the molecule in the mass spectrum.

Finally, simulated fragmentation was performed on all candidate substances in the Transformer prediction list, generating a series of SMILES fragment trees. By comparing the similarity between the SMILES fragment trees of each predicted molecule and the molecular fragment trees generated from tandem mass spectrometry, substance similarity was ranked. The SMILES fragment tree with the highest similarity score is considered to most likely contain the substance under test. The closest subtree to the molecular fragment tree can be found in the highest-scoring SMILES fragment tree, containing structures with the highest similarity to the substance under test. Additionally, the possible dissociation pathways of each fragment peak in the mass spectrum are provided by the highest-scoring SMILES tree, facilitating structural annotation.

Experimental results demonstrate that the multi-type fragment tree similarity scoring mechanism ensures confident ranking of generated results, where higher fragment tree similarity indicates a higher likelihood of the molecule being the substance under test or a part of it, offering a novel method for comparing molecular similarity. For complex or poorly predicted molecules, the model's prediction effectiveness can be enhanced by repeating predictions, thereby improving the model's ability to recognize spectra.

We validate the model's performance on different datasets, especially for predicting the structures of various drugs using miniaturized linear ion trap mass spectrometers. It's worth noting that compared to other mass spectrometry prediction models, for instance, SIRIUS4, which includes CSI: FingerID, demands a higher level of mass resolution (The mass deviation should be within 20 ppm, which can be challenging for miniaturized mass spectrometers), TeFT is the first model designed for miniaturized mass spectrometers. The tolerance of the Transformer model for mass resolution allows us to achieve structure prediction. While this might result in more candidate substances, the inclusion of domain-specific fragmentation rules can effectively aid in simulating the fragmentation of these substances, and then rank them according to the similarity score of two trees. When constructing the training dataset, we retained some highly similar molecules, which did not lead to data leakage. For deep learning models, the input

consists of the spectra of molecules, and structural similarity between molecules does not necessarily imply high similarity in their tandem spectra. For instance, benzoic acid and para-benzoic acid exhibit highly similar molecular structures, but their spectra differ significantly. Similarly, for luteolin and kaempferol, experimental evidence shows that luteolin contains fragmentation peaks at m/z = 153,161,199,213,223, while kaempferol peaks at m/z = 153,165,213,241,258, with only m/z = 153 being identical. Even slight structural differences can lead to significant variations in the tandem spectra of substances. Typically, secondary fragmentation spectra of substances, after excluding miscellaneous peaks, consist of 3–5 fragment peaks. The differences between the two peaks represent considerable variations between spectra. Therefore, to enable the model to better understand the relationship between spectra and molecular structures, we retained similar molecules for training purposes.

While our model may not exhibit exceptional performance advantages in predicting high-quality resolution spectra compared to other deep models, its focus lies in predicting tandem spectra with low resolution and accuracy. This capability enables effective on-site mass spectrum recognition. Models like MSNovelist rely on predicting molecular fingerprints for structure prediction, which imposes high requirements on mass accuracy. Similarly, the MassGenie model is geared towards recognizing spectra with high resolution and accuracy. Therefore, we devised a recognition model tailored for low-resolution spectra, effectively meeting the needs of on-site mass spectrometry and expanding its application scope.

However, this work has limitations, as we only trained TeFT on spectra recorded in positive ion mode (H+), restricting its applicability. Additionally, using low-resolution mass spectrometry to generate fragmentation tree models often results in a higher number of candidate results. We addressed this by limiting the types and numbers of elements, but challenges in finding the correct fragmentation tree still exist.

In other studies, the application of Transformer models in various molecular structure generation tasks, such as reaction prediction[50], has been explored. Reaction prediction is treated as a machine translation problem between SMILES strings representing reactants, reagents, and products. In such cases, using independent multi-head attention molecular transformer models has shown promising results. However, in our study, the Transformer model applied is specifically designed for generating the complete structure of the target substance based on MS$^2$ spectra. The input information for the Transformer consists solely of MS$^2$ spectra, without additional contextual information. Experimental results indicate that the model's predictions for molecular structures are not always unique. To address this, a subsequent fragment tree model and simulated fragment similarity scoring are employed after the Transformer model, enabling the identification of the most similar substances. This approach achieves de novo generation of the structure of the target substance.

Additionally, pre-trained models like ChemBERTa[51] have found extensive applications in tasks such as molecular property prediction, classification, and medicinal chemistry. Through fine-tuning, these models can effectively predict specific downstream tasks, including drug property prediction. However, research on establishing large-scale self-supervised pre-training models for mass spectrometry prediction is limited. Some studies have employed pre-trained models for extracting molecular features, which are subsequently combined with MS/MS datasets to accomplish mass spectrum prediction across multiple datasets[52]. In contrast to those approaches, this study is specifically tailored for predicting mass spectrum data with low-quality resolution in a miniaturized mass spectrometer, building upon dataset predictions. In this study, we adopted a lightweight Transformer architecture to meet the requirements of on-site detection and achieved promising results on a miniaturized mass spectrometer with low resolution. The application scenario of our model demands lightweight requirements, making existing models more suitable for on-site detection compared to pre-trained models. These models meet the practical demands of real-world applications. Additionally, existing models have already demonstrated excellent performance in spectrum prediction tasks. Therefore, it is challenging to ascertain whether pre-trained models offer better

performance in this application scenario. Although the use of pre-training models and fine-tuning techniques is not the primary focus of this research, future studies could further explore the application of molecular pre-training models in the field of mass spectrometry prediction.

## Methods
### Dataset
The training set utilized for the Transformer model was created by compiling open-source spectrometry databases GNPS, HMDB (5.0), and MoNA[53–55]. The training set underwent filtration following predefined criteria, including the imposition of constraints such as limiting the molecular weight to less than 500 Da and permitting only the presence of the ten elements C, H, O, N, P, S, Cl, Br, I, and F, which were suitable for applications of our miniaturized ion trap mass spectrometry. Given that a single molecule can be represented by multiple SMILES strings, the training set standardized the SMILES representations for all molecules and removed duplicate entries. Consequently, the finalized training set encompassed 220,638 distinct molecules, each paired with its corresponding mass spectra in positive ion mode.

### Instrumentation
The drug fragmentation experiment was conducted on a custom-made small linear ion trap mass spectrometer with a continuous atmospheric pressure interface. Sample ions are introduced from the sampling tube into the first vacuum chamber, where they undergo assistance via an ion funnel for transmission. Subsequently, they proceed into the second vacuum chamber through a sampling cone, and they undergo quality analysis within the ion trap. To achieve resonant excitation of ions, a pair of AC voltages with equal magnitude but opposite phases is applied to the electrodes within the ion trap. This voltage is commonly referred to as the auxiliary AC excitation signal. The experimental instrument utilized in this study is a hyperbolic linear ion trap, characterized by hyperbolic size parameters with a radius of x = 4 mm and y = 4.25 mm. The RF voltage and auxiliary AC excitation signal are applied to two pairs of hyperbolic electrodes. For a detailed description of the instrumentation, our previous work provided a more detailed description of it[42–46]. The MS$^2$ spectrum of each substance was measured experimentally, and the top five to six fragmentation peaks and their five nearby peak points were selected as experimental data and input into the prediction model.

### Chemical samples and preprocessing
The chemical samples used in this study included flavanols, stilbenes, flavones, and Rotundine. All the samples were purchased from Aladdin Biochemical Technology Co., Shanghai, China and Macklin Biochemical Technology Co., Shanghai, China. All drugs were diluted in methanol to final concentrations ranging from 1 to 100 mg/L.

In the drug ingredient identification experiment, the Anweiyang capsules used in the experiment were purchased from Huizhou Jiuhui Pharmaceutical Co., Ltd. We employ ultrasound-assisted extraction (UAE) as a crucial technique to extract flavonoids from the drugs[56]. Flavonoids are extracted using a methanol aqueous solution. The process involves grinding the capsule powder and 70% methanol solution is used to perform sonication. An ultrasound process is operating at 180 W for 45 min. Upon completion of the ultrasound process, the solution is filtered through a nylon membrane, which was found that drug residue would remain or be adsorbed on the Polyether sulfone Nylon syringe membrane filter (Φ25 mm, pore size: 0.45 μm, Shanghai ANPEL Laboratory Technologies Inc.) and subsequently subjected to experimentation using miniaturized mass spectrometry.

### The training stage of the transformer model
As illustrated in Fig. 6, our study adopted the Transformer model, distinguished by its encoder-decoder architecture exclusively reliant on attention mechanisms, which captured the overarching interdependence between input MS$^n$ spectral data and output SMILES string. The model comprised six encoder layers and six decoder layers. In the encoder layers, two sub-layers were integrated: one housing the multi-head self-attention mechanism with

 

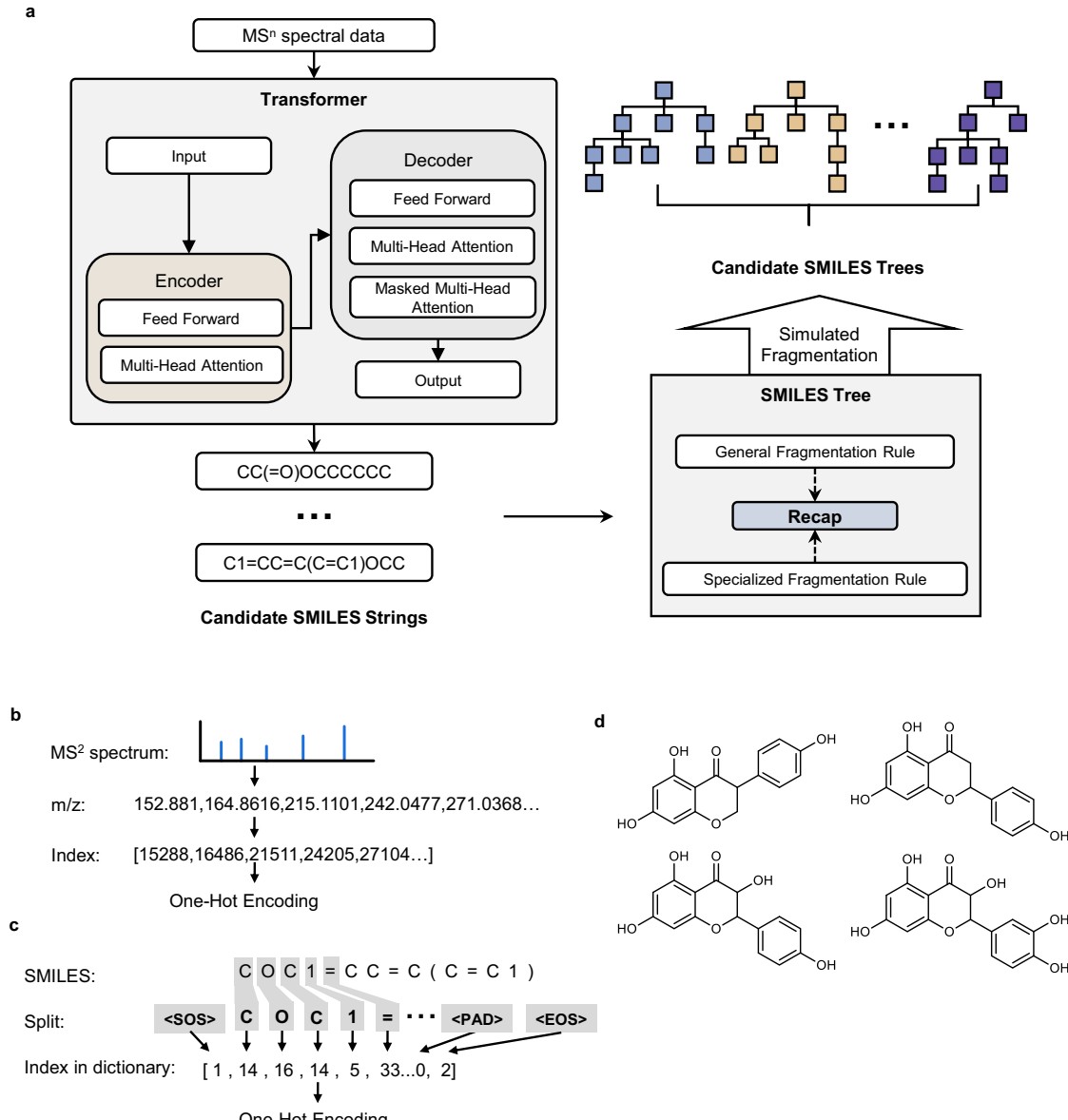

**Fig. 6 | Procedure for data processing and generating SMILES trees following the input of MSn spectrum into a transformer model. a** Detailed architecture of the Transformer and SMILES tree model in TeFT. **b** Convert MS$^n$ spectrum into vectors. **c** The SMILES string is segmented into tokens and subsequently transformed into vectors, with each vector corresponding to the token's index within the dictionary. **d** Four randomly chosen examples of incorrect predictions (top candidate) from the dataset.

eight parallel attention heads, and the other hosting the feed-forward layer. Meanwhile, the decoder layers incorporated three sub-layers. All sub-layers and embedding layers generated output dimensions of $d = 512$.

The input data comprises a peak list, limited to a maximum of 100 peaks selected based on their ion intensities. To transform MS$^n$ spectral data into suitable inputs for the Transformer, initially, we retain solely the mass-to-charge ratio (m/z) data while omitting the abundance data. This step is taken because abundance data is influenced by numerous factors that do not facilitate model learning. Then, the m/z data, preserved in floating-point format, is truncated to two decimal places. Following this, it is multiplied by a factor of 100, transforming it into integer values. This process serves the purpose of upholding the model's training accuracy at 0.01 Da. Abundance data is not directly input into the Transformer model, but that doesn't imply that abundance data is disregarded. Before removing the abundance data, we select several peaks with relatively high abundances from the MS$^2$ spectrum as inputs to the model. Typically, noise does not exhibit such high intensity, and we consider these peaks to be the most likely fragments of the substance. Subsequently, the abundance data is removed. For the encoding

of m/z values, we use truncated values to construct a one-hot matrix. This encoding method effectively preserves the ordering and relationships between numerical values. Supplementary experiments for other numerical encoding methods are provided in Supplementary Note 3. During the model's training stage, it is essential to split the SMILES strings of the molecules. Canonical SMILES strings were divided into a list of distinct atomic types (e.g., C, N, O, P, etc.) and associated connectors (such as "[", "(", "=", etc.). Tokenize individual atoms and connectors to form token sequences, which were standardized to commence with the start token ("<SOS>") and terminate with the end token ("<EOS>"). Additionally, they were padded with the padding token ("<PAD>") to achieve a fixed length of 100. Compose atomic types, connectors, and special tokens into a dictionary comprising 44 distinct elements and identify the corresponding indices of the segmented elements within the dictionary to construct the SMILES vector. Additionally, we compared the impact of two molecular representation methods, SMILES and SELFIES, on the model performance, with results provided in Supplementary Note 2. Finally, both "m/z" vectors and "SMILES" vectors were transformed into one-hot encoded matrices.

https://doi.org/10.1038/s42004-024-01189-0　　　　　　　　　　　　　　　　　　　　　　　　　　　　　　　　　　**Article**

The training was executed using the SGD optimizer, with a batch size of 150 and a learning rate of 0.001. Our model was trained on a machine equipped with an RTX3090 GPU, and the entire training process was completed in 19 h, utilizing 7289MB of RAM. During the testing phase, we compared various decoding methods and ultimately selected greedy decoding to generate the final results. Further details can be found in the Supplementary Note 4.

### The inference stage

During the inference stage, following the same preprocessing and peak selection procedures of the input mass spectrometry data for an unknown substance, the model generates the candidate SMILES strings. The TeFT model, based on the Transformer architecture, generates a series of candidate molecules after making multiple predictions on the same mass spectrum. Unlike the MassGenie model, our model does not produce a unique answer. We speculate that this phenomenon occurs due to limitations imposed by the quality resolution of the input spectrum and constraints on the parameters of the Transformer model. Throughout the testing phase, we conducted 100 iterations of the model, which ultimately resulted in several potential SMILES results. These representations collectively form a SMILES list, which is subsequently sent to the SMILES tree generation model. This model entails an inference process applied to the SMILES list to discern the most plausible molecular structure.

The creation of the SMILES tree model involves implementing the RECAP[57] method in the RDKit toolkit. The RECAP method breaks down molecules into fragments by simulating the process of chemical reactions. In comparison to the traditional RECAP method, the Recap approach used in this study is a customized version. Within the framework of the RECAP method, we specify reaction rules for molecules based on common fragmentation patterns observed in mass spectrometry. By simulating the decomposition of molecules according to these rules, we make informed conjectures about the way molecules undergo fragmentation in the mass spectrometer. Mass spectrometry frequently involves chemical bond dissociation and rearrangement in tandem setups, with the dissociation taking various forms, including homolytic, heterolytic, or hemi-heterolytic, while rearrangement encompasses both breakdown and the re-formation of chemical bonds. This simulation was executed utilizing SMARTS, a reaction representation based on SMILES. SMARTS enables molecular structure transformation by specifying the reaction template, making it applicable for substructure matching and chemical reactions.

The above procedures can generate "Node Tree" data, representing the SMILES tree. In this tree, each node signifies a potential fragment, and the directed edges linking pairs of nodes denote potential mass spectrometry fragmentation losses, annotated with the molecular formulas of the structures lost after fragmentation. Currently, we have achieved partial dissociation and rearrangement for specific chemical bonds commonly found in various substances, such as C-C, C-O, and C-N bonds. Additionally, we have integrated well-documented chemical bond rearrangement reactions like the McLafferty rearrangement[58] and RDA rearrangement[59], which are observed in tandem mass spectrometry. Furthermore, we have expanded the dissociation method database by incorporating specific dissociation rules tailored for flavanols[60] and stilbene[61] substances, facilitating their structural identification in analyses. A full list of dissociation and rearrangement rules adopted in our experiments can be found in Supplementary Table S1. Our method offers the flexibility to incorporate fragmentation rules for various types of substances, rendering the database highly extensible. In the process of fragmentation, the substance searches for matching fragmentation rules within the rule database until no further matching chemical structures are identified. To prevent unlimited program execution times, we have imposed a limitation of 1500 nodes for the total count of SMILES tree nodes.

### Generate fragment tree

Software applications such as SIRIUS utilize high-resolution mass spectrometry to produce FT. The FT generation algorithm encompasses several processes, including molecular formula recognition, molecular formula

**Table 3 | Similarity scoring rules for SMILES tree and fragment tree**

| Object | Event | Score |
|---|---|---|
| Parent | All-atom matching | +6 |
| | Nonhydrogen atom matching | +3 |
| | Mismatching | 0 |
| Fragment | All-atom matching | +5 |
| | Nonhydrogen atom matching | +2 |
| | Mismatching | 0 |
| Loss | All-atom matching | +4 |
| | Nonhydrogen atom matching | +1.5 |
| | Mismatching | 0 |

filtering, and weight calculation. Molecular formula recognition entails computing all potential element combinations within a specific mass deviation range. Subsequently, candidate molecular formulas are subjected to specific filtering rules. It is noteworthy that decreasing the resolution of the mass spectrometry spectrum upsurges the number of potential molecular formulas. To cater to miniaturized linear ion trap mass spectrometers' data processing needs, we have devised an algorithm that builds upon the original method to generate FT from low-resolution mass spectrometry data. This algorithm retains the fundamental FT calculation procedure, and it involves narrowing down potential molecular formulas by restricting the kinds and quantities of elements during molecular formula identification. To address any system errors or measurement inaccuracies encountered in small mass spectrometers, we implement error correction by employing multiple measurement averaging and deconvolution integration techniques on the spectrogram data. The details of the deconvolution method can be found in Supplementary Note 1. The processed spectrogram typically satisfies the criteria for generating more precise FTs.

### Similarity score

This study introduces an extension to the alignment scoring mechanism of traditional FTs that allows for calculating the similarity between SMILES trees and traditional FTs. This method enables the identification of the fragmentation pattern that is closest to the molecular FT in the SMILES tree, thus identifying structures with high similarity to the tested molecule, and achieving substance recognition and spectral structure annotation.

When aligning FTs, the final score of the tree is mainly determined by the matching scores of losses and fragments. During this process, scores are assigned by comparing the similarities and differences in the types and numbers of loss elements between each node's molecular elements and two nodes. The scoring rules are shown in Table 3. In the scoring rules, we have appropriately reduced the score of loss matching to avoid the impact of possible long-chain losses on the overall score.

We standardize the scores obtained by using perfect matching as the denominator. The higher the score, the greater the similarity. We use the similarity score of the FT as a new indicator to evaluate the degree of molecular similarity while maintaining high credibility.

### Molecular re-prediction

The SMILES tree with the highest similarity score often provides crucial structural information, and when representing molecules using SMILES strings, it is possible to arbitrarily specify the starting atom. This feature allows us to make supplementary predictions based on the SMILES strings of fragments when the model's overall structure prediction for the substance is less accurate, thereby substantially improving the model's molecular recognition capabilities. During the iterative prediction phase, we utilize SMILES fragment strings. The model can predict and extend the structure at the end of the sequence while preserving the integrity of the fragment structure. Details of the repeated predictions have been added to Supplementary Note 5.

## Data availability

The experimental data from the public database used in this study can be downloaded from the following website. GNPS dataset: https://gnps-external.ucsd.edu/gnpslibrary/ALL_GNPS.json, HMDB 5.0: https://hmdb.ca/downloads, MoNA: https://mona.fiehnlab.ucdavis.edu/downloads. Filtered public data used for training and evaluating the TeFT model can be downloaded alongside our code included at https://github.com/thumingo/TeFT.git. In the article, the original data for Fig. 2 is provided in Supplementary Data 1. The detailed data for Fig. 3 and Table 2 in the main text is provided in Supplementary Data 2. In Supplementary Information, the detailed data for Supplementary Fig. S1 and Supplementary Fig. S4 is available in Supplementary Data 3 and Supplementary Data 4. Supplementary Data 1–4 are located in the file "Supplementary Data.xlsx".

## Code availability

TeFT is available on GitHub (https://github.com/thumingo/TeFT.git). The model, evaluation code and train code are implemented in Pytorch, version 1.12.0 on Python 3.7.13. The version of RDKit is 2020.09.1.0.

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

## Acknowledgements

This work was supported by the National Natural Science Foundation of China (No. 22374164) and the Shenzhen Natural Science Foundation (No. JCYJ20200109142824889 and No. RCBS20210609104339043).

## Author contributions

Yiming Yang: Investigation, methodology, algorithm design, writing—original draft preparation, data processing. Shuang Sun: Resources. Shuyuan Yang: Algorithm design, resources. Qin Yang: Resources. Xinqiong Lu: Resources. Xiaohao Wang, Quan Yu, Xinming Huo: Supervision. Xiang Qian: Writing—review & editing, methodology, conceptualization, and supervision.

## Competing interests

The authors declare no competing interests.
