## [Peer Review File · Communications Chemistry]

Reviewers' comments:

Reviewer #1 (Remarks to the Author):

Strengths:

The authors have introduced a novel method that fuses a transformer model with a manually constructed fragmentation tree to establish a framework for predicting mass spectrometry (MS) fragments. Notably, the model demonstrates the capacity to generate accurate fragments for chemical compounds it has not previously encountered, showcasing its potential for generalization.

Weakness

1. Experimental Rigor and Benchmarks:

The current strategy for data splitting is random, which does not account for chemical similarity between the training and test datasets. This raises the concern that the sets may contain closely related compounds, differing only slightly, such as a methyl group being replaced by an ethyl group. It is suggested that a chemical diversity split be implemented. This could involve ensuring that the maximum Tanimoto similarity between any training and test compound is below 0.5, thereby minimizing data leakage.

A thorough benchmarking study is recommended to evaluate the model against existing deep learning and machine learning methods like CANNOPUS, MassGenie, DarkNPS, and MSNNoverlist, as well as established approaches like XRank and SIRIUS. This would entail retraining those models with the proposed data split and comparing the results to understand the unique advantages of the presented model and guide researchers in choosing the most suitable method for MS annotation.

2. Model Architecture:

There is potential in exploring the transformer model's ability to predict valid fragments without the support of hand-crafted fragmentation rules. An intriguing reference is a study where a transformer successfully predicted compound building blocks without predefined reaction rules. (ref paper <https://pubs.acs.org/doi/10.1021/acscentsci.9b00576>) The authors could enhance their findings by conducting experiments with and without fragmentation tree (FT) guidance to determine its impact and possibly strengthen the argument for the necessity of FT.

3. Technical Refinements:

- Standardizing on InChIKeys rather than SMILES could improve consistency and interoperability.
- Undertaking hyperparameter tuning could reveal whether the model might maintain or even enhance its performance with a more streamlined parameter set, potentially increasing the efficiency of the model.
- It could be beneficial to experiment with pretrained models like ChemBERTa transformers to see how they fare on this task compared to the authors' custom transformer architecture, either by fine-tuning them or starting from scratch.

- A more detailed description of the data processing steps is essential to assess the model's generalizability. Publishing the data processing and model training code would contribute to the reproducibility and transparency of the research.

Conclusion:

The manuscript is currently not recommended for acceptance due to a lack of comprehensive experimentation to substantiate the model's efficacy beyond the scope of overfitting. Should the authors conduct the suggested rigorous experiments and benchmarks, achieving state-of-the-art performance, the manuscript would likely hold greater significance within the scientific community.

Reviewer #2 (Remarks to the Author):

The authors propose a Transformer Fragment Tree (TeFT) methodology that can characterize compounds' molecular structures starting from low-resolution mass spectrometry data. After a comparative evaluation with existing methodologies on public datasets, they conducted experiments using a miniaturized linear ion trap mass spectrometer.

Initially, they evaluate the approach on 23 commercially available compounds, showing the methodology's performance in recognizing known substances.

Finally, they test the model's applicability on a Chinese medicine substance called the Anweiyang capsule, showing that the best-scoring substance predicted by TeFT is a compound reportedly present in the studied medication.

The methodology is interesting, and the results presented support the paper's claims. However, further experiments and more detailed evaluation are required to strengthen the overall quality of the paper and make it suitable for publication.

Comments/suggestions ordered according to the paper structure follow.

On page 2, the authors report a series of prior works using machine learning and deep learning that address similar problems in structural characterization from MS data. However, none of these methods appear in the public dataset evaluation from the initial results section (only SIRIUS4 appears as a baseline). While it is understandable that a complete adaptation of all the models to the problem considered is challenging, it would be interesting to consider some of those in these experiments (pick one or two) to replace the transformers component of TeFT to predict the SMILES list inputted to build the trees with Recap.

Another note on the related art: the authors mention that MassGenie presents a complicated training process due to its 400+ million nodes. If by nodes the authors mean parameters, they should consider revising the statement as this model size is standard, and relatively small GPU accelerators can handle such size.

In the comparison with SIRIUS4 on page 4, the authors discuss the overall prediction accuracy of TeFT.

They state it is superior to SIRIUS4 based on qualitative observations of the reported similarity plots. To support this statement quantitatively, I recommend conducting a non-parametrical statistical test (e.g., Wilcoxon signed rank) on both similarity plots and reporting their p-values. Also, as the authors speak about the accuracy, I suggest including a table or line plot of the top-k methods' accuracy with k varying from 1 to 5.

In evaluating TeFT on the 23 commercially available compounds purchased (page 5), consider formatting the table to report the minimum/maximum similarities and number of correct substances as separate columns. Currently, the format could be more straightforward. Regarding the ranking, one might consider having a rank distribution plot for each of the four substance types in supplementary.

Regarding training the transformers model (page 9), some modeling choices will benefit from a more in-depth analysis and motivation.

It needs to be clarified how the one-hot encoding of the truncated m/z values is implemented. Is the truncated value used as-is? Doesn't this translate into unnecessarily large embeddings? I recommend elaborating on whether this approach preserves ordering between numbers.

Moreover, there have been attempts to encode the numerical value in transformer models, and it would be interesting to perform some ablation studies comparing the proposed approach with other techniques used in the literature [1, 2].

Regarding the smiles embedding, it is more a curiosity than a comment. Why one-hot also here? In literature, there are plenty of examples relying on learned embeddings. This strategy usually guarantees to be able to work with smaller embedding matrices (usually, embedding size is limited to 8/16/32 dimensions, generally smaller than the one resulting from one-hot).

On page 10, some clarifications are needed when discussing the inference stage. When the authors refer to 100 iterations, do they refer to 100 forward passes? If yes, what is the batch size?

Here I assume the decoding strategy used is greedy decoding. Is it? If yes, a beam search would allow quickly generating multiple compounds in a single pass as a single sample could decode multiple molecules.

In the same section, I would invite the authors to comment and elaborate more on the reason behind the expectation of observing increased prediction variability with lightweight models (i.e., "However, this lightweight configuration also contributes to the nonuniqueness of the model's output".) as this statement has no theoretical foundation unless I am missing something.

On page 11, please include a citation to the original Recap paper [3].

On page 12, I found the molecular re-prediction aspect interesting. It is worth including examples of how this impacts the model performance by providing actual cases the authors observed. They can be included in the supplementary material.

Always on page 12 regarding the code availability. I genuinely appreciate the effort for open-sourcing the code, but there are no instructions for environment setup, which hinders reproducibility.

Ensure to include a "requirements.txt" file reporting all dependencies with recommended versions needed to run the code.

By browsing the repo, I found many dependencies a user needs to guess, which we cannot expect when putting code out (e.g., scipy, scikit-image, matplotlib).

References:

[1] <https://doi.org/10.1038/s42256-023-00639-z>

[2] <https://doi.org/10.48550/arXiv.2310.02989>

[3] <https://doi.org/10.1021/ci970429i>

Reviewer #3 (Remarks to the Author):

This study proposes a novel structure elucidation scheme from tandem mass spectra. The proposed method consists of a compound prediction by the Transformer model and fragmentation per expert knowledge rules. The predicted compounds by the Transformer model are ranked based on the matching score between the decomposition tree from the compound and the FT about molecular weight. The study was well planned, and their findings are interesting to experimental chemists. There are several comments that I noticed when reading the manuscript as follows.

1. SMARTS fragmentation list. Evaluation of the SMARTS list for simulated fragmentation seems necessary, including the coverage of the rules. Did the modified RECAP (retrosynthetic combinatorial analysis procedure) rules cover fragmentation rules in mass spectroscopy? Or, is "Recap" different from RECAP?
2. MS2 selection dependence. Is the proposed prediction scheme irrelevant to the selection of MS1 peaks for MS2 decomposition? If this is not the case, a small description of collecting data would be necessary.
3. Validated compounds. Did the authors confirm that the validated compounds in this study were not included in the data set for Transformer training: 220,638 compounds?
4. Noise of MS. Removing the abundance data drastically simplifies the prediction task while increasing the possibility of selecting peaks with noises, in particular, low-resolution MS. Unlike the authors' introducing filtering technique for generating TS, the input of the Transformer seems original spectra without intensity. Can the noise be ignored overall?
5. Single compound prediction. Can the proposed approach be employed for the structure identification of a mixture of compounds? Or, is this approach solely applicable to elucidate a pure compound after column chromatography for example? Since the Transformer model seems to generate a single compound, which compound is generated when inputting a spectrum of a mixture? As shown in the manuscript, one of the main ingredients of Anweiyang Capsule was retrieved, however, what are the other ingredients? Did the Transformer model generate chemical-structurally different SMILES strings as candidates for the other ingredients? The removal of the abundance data might make the prediction of the main compound prediction difficult.
6. If several compounds are matching the same input of the Transformer model, does the Transformer model generate all the compounds as input for training?

We sincerely appreciate the time and effort the reviewing team dedicated to evaluating our manuscript. Their valuable insights and suggestions are highly regarded by our team. We recognize the significance of their professional knowledge and expertise, which greatly contributes to the refinement of our research. In response to the questions and suggestions raised by the reviewing team, we will now address each of them point-by-point carefully.

Responses to Reviewer #1

1. Experimental Rigor and Benchmarks:

The current strategy for data splitting is random, which does not account for chemical similarity between the training and test datasets. This raises the concern that the sets may contain closely related compounds, differing only slightly, such as a methyl group being replaced by an ethyl group. It is suggested that a chemical diversity split be implemented. This could involve ensuring that the maximum Tanimoto similarity between any training and test compound is below 0.5, thereby minimizing data leakage.

A thorough benchmarking study is recommended to evaluate the model against existing deep learning and machine learning methods like CANNOPUS, MassGenie, DarkNPS, and MSNovelist, as well as established approaches like XRank and SIRIUS. This would entail retraining those models with the proposed data split and comparing the results to understand the unique advantages of the presented model and guide researchers in choosing the most suitable method for MS annotation.

Response: Regarding the concerns raised about proposed data-splitting strategy, we understand the apprehension about data leakage in deep learning models. We respect and carefully consider the opinions, however, in this study, the Transformer model is employed to process tandem mass spectrometry (MS²) data as input, generating corresponding molecular SMILES representations as output. It's worth noting that even subtle structural differences in molecules can lead to significant differences in MS². Therefore, during the dataset-splitting process, there is no necessity to ensure absolute dissimilarity between molecules in the training and test sets. The current training process ensures no identical molecules between the training and test sets, minimizing the risk of data leakage. Thus, we believe the current data-splitting strategy meets the requirements for the model's operation.

Simultaneously, addressing the concerns about experimental rigor and benchmarks, we appreciate the identification of limitations in our model's performance comparison and we have made the modifications according to the suggestions. To address this, we expanded the comparison beyond the benchmark method SIRIUS4, incorporating a comparison between the TeFT

model and another evaluation method, MetFrag. MetFrag is an effective mass spectrometry database query method, and we tested it on the validation set and the CASMI 2017 challenge dataset for a comprehensive performance evaluation. The results of these tests indicate that, compared to commonly used database query methods, the Transformer model exhibits superior predictive ability, producing more reasonable SMILES structures or providing valuable references from mass spectrometry. We believe the comparison with multiple benchmark models can attest to the excellent performance of the TeFT model.

2. Model Architecture:

There is potential in exploring the transformer model's ability to predict valid fragments without the support of hand-crafted fragmentation rules. An intriguing reference is a study where a transformer successfully predicted compound building blocks without predefined reaction rules. (ref paper <https://pubs.acs.org/doi/10.1021/acscentsci.9b00576>) The authors could enhance their findings by conducting experiments with and without fragmentation tree (FT) guidance to determine its impact and possibly strengthen the argument for the necessity of FT.

Response: In response to the questions about the model architecture and the Transformer model referenced in another study on chemical applications, it's crucial to highlight that our research differs from the referenced study. The chosen model architecture is necessary for our specific task. The referenced study applied the Transformer model to chemical reaction prediction, treating it as a machine translation problem between SMILES strings of reactants, reagents, and products. In that context, using an independent multi-head attention molecular transformer model demonstrated good results. However, our study deviates significantly as we focus solely on generating complete structures of target substances based on fragment mass spectra. The input information for the deep learning model consists only of MS2 spectra, without additional contextual information. Experimental results indicate that the model's predictions of molecular structures are not unique in certain situations. Therefore, subsequent fragment tree models and simulated fragment scoring are needed to identify the most similar substances.

3. Technical Refinements:

- Standardizing on InChIKeys rather than SMILES could improve consistency and interoperability.

Response: In addition, in response to another issue with our molecular representation, specifically the limitation of SMILES string representation. We apologize for not explicitly stating the reasons for choosing this representation in the paper. Due to the inherent instability of SMILES strings, a certain proportion of molecular structures generated using SMILES is considered

invalid. While the InChIKeys representation, as mentioned, provides a consistent representation of molecules, it lacks inherent meaning and relies heavily on related databases for interpretation, significantly limiting its application in deep learning models. Given the limitations of InChIKeys and SMILES, we chose to conduct further experiments using an alternative molecular representation called SELFIES. This representation exhibits higher fault tolerance and robustness. Results are available in the supplementary materials, showing that, in terms of stability, SELFIES outperforms SMILES; however, the similarity of the final generated results is slightly lower compared to SMILES (similarity=1). The primary focus of this study is on combining deep learning models with subsequent fragment tree models for reasonable inference of target substances. The choice of representation method has no significant impact on experimental results. In conclusion, this study adopts SMILES strings for molecular representation as it overall suits our objectives.

- Undertaking hyperparameter tuning could reveal whether the model might maintain or even enhance its performance with a more streamlined parameter set, potentially increasing the efficiency of the model.

- A more detailed description of the data processing steps is essential to assess the model's generalizability. Publishing the data processing and model training code would contribute to the reproducibility and transparency of the research.

Response: Upon submitting the article, we overlooked providing details on the model parameter selection process and the absence of data processing and model training in the uploaded code. We appreciate the prompt identification of this issue. Detailed information about parameter tuning during the model training process and its final impact is provided in the supplementary materials, confirming that the current parameters yield optimal performance. Additionally, we have supplemented the relevant code on GitHub.

- It could be beneficial to experiment with pretrained models like ChemBERTa transformers to see how they fare on this task compared to the authors' custom transformer architecture, either by fine-tuning them or starting from scratch.

Response: The feedback also includes a discussion on pre-trained models. We understand the keen interest in pre-trained models due to their strong performance, with models like ChemBERTa widely applied in tasks such as molecular property prediction, classification, and medicinal chemistry. However, it's important to note that research on establishing large-scale self-supervised pre-trained models in the field of mass spectrometry prediction is limited. In this study, we aimed to reduce model parameters to enhance performance on low-resolution miniaturized mass spectrometry platforms. To achieve this, we adopted a lightweight Transformer architecture. It's noteworthy that the use of pre-trained models and fine-tuning is not the primary focus of this research.

Future studies can further explore the application of molecular pre-trained models in the field of mass spectrometry prediction to improve predictive capabilities.

Responses to Reviewer #2

On page 2, the authors report a series of prior works using machine learning and deep learning that address similar problems in structural characterization from MS data. However, none of these methods appear in the public dataset evaluation from the initial results section (only SIRIUS4 appears as a baseline). While it is understandable that a complete adaptation of all the models to the problem considered is challenging, it would be interesting to consider some of those in these experiments (pick one or two) to replace the transformers component of TeFT to predict the SMILES list inputted to build the trees with Recap.

Response: Regarding the meaningful suggestion about using benchmark methods to replace the Transformer module in TeFT, predicting input SMILES lists to construct a tree using Recap, we find this suggestion very practical and have supplemented relevant experiments in the manuscript. We added MetFrag test results for comparison with the baseline. Additionally, we replaced the original scoring mechanism with simulated fragments, constructed fragment trees, and evaluated their similarity to SIRIUS and MetFrag predictions. We observed that among substances ranked first in the original SIRIUS score, 56% also ranked first in the fragment tree score, while MetFrag had a proportion of 16%. Furthermore, SIRIUS retrieved substances from the fragment tree score list totaling 27.9%, while MetFrag was 26.7% (ranked first). The Transformer model demonstrates higher flexibility in generating similar molecules, and the fragment tree similarity scores also indicate a certain level of reliability. It's important to note that only the TeFT model is feasible for low-resolution spectra.

Another note on the related art: the authors mention that MassGenie presents a complicated training process due to its 400+ million nodes. If by nodes the authors mean parameters, they should consider revising the statement as this model size is standard, and relatively small GPU accelerators can handle such size.

Response: About the model size and processor performance, we apologize for any confusion caused by the unclear wording in our paper. There was a misstatement in our original text, which has been corrected in the revised version. The MassGenie article pointed out, "We should point out that this success did require what for an academic biology laboratory is a substantial computing resource—the DGX A100 8-GPU system we used to allow us to train a 400-million node network, which would have represented a world record for

the largest published deep learning network less than three years ago." Our experimental conditions make large-scale model training similar to MassGenie challenging, and our ultimate goal is to achieve a miniaturized MS system for on-site applications. Therefore, we simplified the model parameters to make it feasible for GPU operation with slightly lower specifications.

In the comparison with SIRIUS4 on page 4, the authors discuss the overall prediction accuracy of TeFT. They state it is superior to SIRIUS4 based on qualitative observations of the reported similarity plots. To support this statement quantitatively, I recommend conducting a non-parametrical statistical test (e.g., Wilcoxon signed rank) on both similarity plots and reporting their p-values. Also, as the authors speak about the accuracy, I suggest including a table or line plot of the top-k methods' accuracy with k varying from 1 to 5.

Response: As for the issue of model performance comparison, we acknowledge the necessity of addressing the problem, and we have added relevant experiments. Initially, we used the Wilcoxon signed-rank test for non-parametric statistical analysis, and the final p-values are reported in the table. These results indicate significant differences in test performance compared to the SIRIUS model and MetFrag testing method (we added an established comparison method). Additionally, in the supplementary materials, we provide the top five accuracy results for both benchmark methods and the TeFT model.

In evaluating TeFT on the 23 commercially available compounds purchased (page 5), consider formatting the table to report the minimum/maximum similarities and number of correct substances as separate columns. Currently, the format could be more straightforward. Regarding the ranking, one might consider having a rank distribution plot for each of the four substance types in supplementary.

Response: The suggestions for modifying the presentation of experimental results in our tables were noted. We found that the table indeed lacked clarity and has now been revised. We also added a figure describing the ranking of the most similar candidate structures after performing fragment tree similarity scoring.

Regarding training the transformers model (page 9), some modeling choices will benefit from a more in-depth analysis and motivation.

Response: As mentioned earlier, the model parameter selection has been supplemented in the supplementary materials, demonstrating the rationality of the current model's selected parameters.

It needs to be clarified how the one-hot encoding of the truncated m/z values is implemented. Is the truncated value used as-is? Doesn't this translate into

unnecessarily large embeddings? I recommend elaborating on whether this approach preserves ordering between numbers.

Moreover, there have been attempts to encode the numerical value in transformer models, and it would be interesting to perform some ablation studies comparing the proposed approach with other techniques used in the literature.

Response: In response to the potential shortcomings in our current model's m/z value encoding. Currently, for m/z value encoding, we use truncated values to construct a one-hot matrix. This encoding method effectively preserves the ordering and relationships between numerical values. During the dataset creation process, substances with a relative molecular mass greater than 500 were excluded to reduce the dimensionality of the one-hot matrix. Although the final dimension of this one-hot matrix is still relatively large, it is considered necessary for the model training process. We understand the question about the encoding method, and thus, we conducted a series of experiments changing the numerical encoding method. With the reference in the supplementary materials, we used Vaswani's definition of numerical encoding to train the Transformer model. Unfortunately, adopting this encoding method did not improve the model's convergence; instead, it tended to diverge. The change in the CrossEntropyLoss values indicates a significant negative impact of altering the numerical encoding method on the model. Therefore, we adhere to the original numerical encoding method and are open to exploring more reasonable encoding methods in future research.

Regarding the smiles embedding, it is more a curiosity than a comment. Why one-hot also here? In literature, there are plenty of examples relying on learned embeddings. This strategy usually guarantees to be able to work with smaller embedding matrices (usually, embedding size is limited to 8/16/32 dimensions, generally smaller than the one resulting from one-hot).

Response: In response to the encoding of SMILES strings. We would like to clarify that the complete decomposition followed by one-hot encoding is a commonly employed technique in many studies. This encoding method involves splitting SMILES strings into individual letters, symbols, or numbers for convenient processing. It allows this encoding format to represent any substance without being constrained by predefined vocabularies. In this research, the objective is to generate molecular structures from scratch, and the adoption of this segmentation method enables the model to generate candidate molecular structures without limitations.

On page 10, some clarifications are needed when discussing the inference stage. When the authors refer to 100 iterations, do they refer to 100 forward passes? If yes, what is the batch size?

Response: In the inference stage, we appreciate the inquiry that highlighted a lack of clarity in our expression. When we mentioned 100 iterations, we referred to 100 forward passes, with the batch size typically determined by the number of test substances, commonly set to 1.

Here I assume the decoding strategy used is greedy decoding. Is it? If yes, a beam search would allow quickly generating multiple compounds in a single pass as a single sample could decode multiple molecules.

Response: For the decoding strategy, we acknowledge the deficiency in discussing the decoding strategy. The model adopts a greedy decoding strategy, and we can view this strategy as a special case of beam search decoding. For the sake of experimental completeness, we supplemented the study with beam search decoding and refined the code accordingly. Experimental data indicates that the beam width does not significantly impact the final results. The model's output consistently tends to resemble a few molecules closely related to those generated by the greedy decoding method. Furthermore, an analysis of the final output suggests that although beam search decoding can generate more diverse molecules, there is no significant improvement in similarity.

In the same section, I would invite the authors to comment and elaborate more on the reason behind the expectation of observing increased prediction variability with lightweight models (i.e., "However, this lightweight configuration also contributes to the nonuniqueness of the model's output".) as this statement has no theoretical foundation unless I am missing something.

Response: Additionally, the feedback mentioned a misrepresentation in the description of the lightweight model in the article. We sincerely apologize for any lack of clarity in our expression. The intended meaning is to draw a comparison with MassGenie, highlighting that our model, based on the Transformer architecture, yields a series of candidate molecules upon multiple predictions of the same mass spectrum. In the MassGenie article, it is mentioned that the MassGenie model "seems to act as a Las Vegas algorithm, in that it either gives the correct answer or simply states that it cannot find one." We speculate that this phenomenon occurs due to limitations in the quality resolution of the input spectrum and the parameter constraints of the Transformer model. However, it's important to note that in some network architectures other than Transformer, a lightweight design does not necessarily imply non-unique model outputs.

On page 11, please include a citation to the original Recap paper [3].
On page 12, I found the molecular re-prediction aspect interesting. It is worth including examples of how this impacts the model performance by providing

actual cases the authors observed. They can be included in the supplementary material.

Always on page 12 regarding the code availability. I genuinely appreciate the effort for open-sourcing the code, but there are no instructions for environment setup, which hinders reproducibility.

Ensure to include a "requirements.txt" file reporting all dependencies with recommended versions needed to run the code.

By browsing the repo, I found many dependencies a user needs to guess, which we cannot expect when putting code out (e.g., scipy, scikit-image, matplotlib).

Response: We acknowledge the feedback for bringing to our attention the incompleteness of our material. We have addressed this concern by adding supplementary information to the main text regarding the Recap reference and including details about the model's repetitive predictions in the supplementary materials. The necessary information about the model's training environment and dependencies has also been uploaded to our GitHub repository.

Responses to Reviewer #3

1. SMARTS fragmentation list. Evaluation of the SMARTS list for simulated fragmentation seems necessary, including the coverage of the rules. Did the modified RECAP (retrosynthetic combinatorial analysis procedure) rules cover fragmentation rules in mass spectroscopy? Or, is "Recap" different from RECAP?

Response: We apologize for not providing a more detailed explanation in the original manuscript about the Recap method we employed. It's crucial to note that the Recap method we adopted strives to encompass most fragmentation rules that may occur in mass spectrometry, such as the cleavage of chemical bonds like C-C, C-O, and C-N, as well as McLafferty and RDA rearrangements. Additionally, we have included specific types of compounds that may produce characteristic fragments, such as flavonoid compounds and glycosides. It's worth mentioning that our fragment library is an extensible database, allowing the addition of corresponding substance fragment rules at any time. In comparison to the standard Recap method, the Recap method used in this study can be considered a customized version. Within the Recap framework, we specified molecular reaction rules based on common fragmentation patterns in mass spectrometry. By simulating the molecular breakdown according to these rules, we made informed guesses about how molecules fragment in the mass spectrometer.

2. MS2 selection dependence. Is the proposed prediction scheme irrelevant to the selection of MS1 peaks for MS2 decomposition? If this is not the case, a small description of collecting data would be necessary.

Response: Also, for the opinions about the selection of MS2 peaks, we chose pure samples (or extracted analytes from simple pre-treatment processes) for experimentation. During the experimental process, we selected the MS1 peak with the highest abundance, representing the molecular ion peak of the target substance. Subsequently, we isolated and fragmented this peak to obtain the MS2 spectrum.

3. Validated compounds. Did the authors confirm that the validated compounds in this study were not included in the data set for Transformer training: 220,638 compounds?

Response: We can ascertain that the substances used in the testing phase do not appear in the training set. To achieve this, we subjected the SMILES representations in the dataset to normalization, eliminating all duplicate molecules. Subsequently, a subset of substances was randomly selected to form the test set, ensuring non-redundancy with the training set.

4. Noise of MS. Removing the abundance data drastically simplifies the prediction task while increasing the possibility of selecting peaks with noises, in particular, low-resolution MS. Unlike the authors' introducing filtering technique for generating TS, the input of the Transformer seems original spectra without intensity. Can the noise be ignored overall?

Response: Concerning the choice of abundance data, we appreciate the different perspectives they presented. We fully understand the importance of abundance data in mass spectrometry. In this study, although abundance data is not directly input into the Transformer model, it doesn't mean it is entirely overlooked. Before removing abundance data, we select several high-intensity peaks from the MS2 spectrum as input to the model. Generally, noise doesn't exhibit such high intensity, and we believe these peaks are most likely fragments of the substance. Subsequently, abundance data is removed before being input into the model.

5. Single compound prediction. Can the proposed approach be employed for the structure identification of a mixture of compounds? Or, is this approach solely applicable to elucidate a pure compound after column chromatography for example? Since the Transformer model seems to generate a single compound, which compound is generated when inputting a spectrum of a mixture?

Response: In response to the doubts about the substances for which the model is applied, the current model is specifically designed to elucidate individual compounds, typically isolating and fragmenting individual peaks in complex mass spectra to infer potential compounds associated with that peak. It is unable to identify complex spectra of mixtures.

As shown in the manuscript, one of the main ingredients of Anweiyang Capsule was retrieved, however, what are the other ingredients? Did the Transformer model generate chemical-structurally different SMILES strings as candidates for the other ingredients? The removal of the abundance data might make the prediction of the main compound prediction difficult.

Response: The meaningful question raised about the ingredient identification experiment with Anweiyang Capsules highlighted the incompleteness of our experimental discussion. We identified the structure of one of the main components of the capsule in the experiment, supported by literature, but lacked discussion on other components. For other components, the TeFT model can provide a series of predicted molecular structures, and we have incorporated them into the main text. Since there is a lack of clear evidence in other literature and existing resources to confirm the substances corresponding to the remaining peaks, the predicted structures are provided for reference purposes only. We compared the TeFT model predictions based on the miniaturized MS method with the test results from commercial high-resolution MS (Thermo Fisher Q-Extractive Orbitrap) using the same extracted solution. The molecular formulas of the two structures we predicted match those from commercial high-resolution MS results, providing some evidence of the accuracy of our model's predictions. Our predictive results demonstrate molecular formulas that align with measurements obtained from high-resolution mass spectrometry with very high mass accuracy (the errors are less than 0.01). For the peak at $m/z=262$, our predictive model calculates a theoretical molecular mass of 262.191 ($C_{15}H_{24}N_3O^+$), which precisely matches the high-resolution measurement result of 262.191. The peak at $m/z=249$ exhibits comparable predictive accuracy, with the predicted molecular formula being $C_{16}H_{13}N_2O^+$ and a theoretical molecular weight of 249.102, which closely corresponds to the high-resolution measurement result of 249.111. Because that even small variations of m/z value can indicate totally different molecular formulas, high-resolution MS peaks can be used to determine the molecular formulas of the analytes in the practical applications. This to some extent corroborates the accuracy of our model predictions based on low resolution MS data. However, fully structural accuracies of the predicted molecules should be further verified using multiple analytical methods in the future.

6. If several compounds are matching the same input of the Transformer model, does the Transformer model generate all the compounds as input for training?

Response: Regarding the issue of generating the same output for the same input, the Transformer model can indeed generate a series of candidate substances, including various types of compounds that may correspond to the given spectrum. However, when two substances exhibit the same fragmentation spectrum, the TeFT model may struggle to distinguish between them. In such cases, providing additional information is still necessary for accurate predictions.

REVIEWERS' COMMENTS:

Reviewer #1 (Remarks to the Author):

The authors have made a commendable effort to incorporate some of the feedback received by adding more results and details to their manuscript.

However, there appears to be a need for further clarification regarding the handcrafted rule's performance superiority. While the authors have addressed the comparison with pure transformer models by highlighting the inherent variability and generative nature of these models as an advantage, it might be beneficial to explore this aspect more thoroughly. Additionally, the comparison with pretrained models was not fully addressed. Given the transformative potential of fine-tuning state-of-the-art pretrained models on downstream tasks, particularly when employing transformer models, it is crucial for the authors to demonstrate the benefits of developing a smaller transformer model over leveraging existing pretrained models.

Regarding the issue of data leakage, the authors' response suggests that subtle structural differences in molecules can have a significant impact on MS2 outcomes. While this is an important consideration, it may be useful to provide further evidence or analysis to support this claim, especially since significant differences between highly similar molecules are often considered rare. As the MS will chunk the molecule into small fragments, many fragments might stay the same, and only one or two fragments that have small changes will only lead to small changes in one or two peaks.

Lastly, the benchmark results against other deep learning methods in the same tasks suggest that there may be room for improvement in demonstrating a clear performance advantage. Hopefully, these points can be addressed further to strengthen the manuscript and its contributions to the field.

Reviewer #2 (Remarks to the Author):

The authors satisfactorily addressed my comments, and I believe that the current version of the manuscript is ready for acceptance.

A comment, mainly to guide potential extensions of the work.

My observation regarding the SMILES embeddings has been misunderstood/was unclear.

Leveraging learned embeddings of smaller dimensionality does not alter the vocabulary size (hence, no additional vocabulary constraints compared to one-hot encoding).

Adopting such a strategy can be helpful in these "edge"/on-device applications where model size needs to be compressed as much as possible, as smaller embeddings would allow for smaller models.

I thank the authors for taking the time to incorporate the feedback and substantially improving the manuscript.

Reviewer #3 (Remarks to the Author):

The authors have clarified all my questions and concerns. I confirm that this manuscript is of an acceptable scientific standard.

We sincerely appreciate the time and effort the reviewing team dedicated to evaluating our manuscript. Their valuable insights and suggestions are highly regarded by our team. We recognize the significance of their professional knowledge and expertise, which greatly contributes to the refinement of our research. In response to the questions and suggestions raised by the reviewing team, we will now address each of them point-by-point carefully.

Responses to Reviewer #1

1. However, there appears to be a need for further clarification regarding the handcrafted rule's performance superiority. While the authors have addressed the comparison with pure transformer models by highlighting the inherent variability and generative nature of these models as an advantage, it might be beneficial to explore this aspect more thoroughly.

Response:

We acknowledge the deficiencies in the content of the manuscript. In the final draft, we have strengthened the description of the rule-based embedding model to address the reviewers' concerns.

We adopted a rule-embedded Transformer model. Experimental results from on-site mass spectrometry indicate that due to the limitations of spectral resolution, mass accuracy, and model architecture, the Transformer model does not yield unique output for the same input. During several different inference processes, we may obtain several similar but not entirely identical molecular representations. These molecules have very close molecular weights and highly similar structures. Therefore, further filtering is required to eliminate this "ambiguity." To enable similarity ranking of candidate substances, this study established a similarity scoring mechanism using both molecular fragment tree and SMILES fragment tree models.

By embedding the fragmentation rules of substances, we can identify the most similar substance structures from a series of results predicted by the model, thus completing substance screening. The specific details have been added to the discussion section.

2. Additionally, the comparison with pretrained models was not fully addressed. Given the transformative potential of fine-tuning state-of-the-art pretrained models on downstream tasks, particularly when employing transformer models, it is crucial for the authors to demonstrate the benefits of developing a smaller transformer model over leveraging existing pretrained models.

Response:

We highly concur with the reviewers' perspective on pre-trained models, as they undeniably exhibit advanced predictive performance. Pre-trained models like ChemBERTa have indeed found widespread application across various

domains, including molecular property prediction, classification, and medicinal chemistry. They demonstrate remarkable performance through fine-tuning, particularly in predicting specific downstream tasks such as drug properties. However, the realm of establishing large-scale self-supervised pre-training models specifically for mass spectrometry prediction remains relatively unexplored.

Some studies have leveraged pre-trained models to extract molecular features, which are subsequently integrated with MS/MS datasets to predict mass spectra across diverse datasets. In contrast, our study focuses on predicting mass spectrum data with low-quality resolution in a miniaturized mass spectrometer, relying on dataset predictions. To meet the demands of on-site detection, we adopted a lightweight Transformer architecture and achieved promising results on a miniaturized mass spectrometer with low resolution.

The unique application scenario of our model necessitates lightweight requirements, making existing models more suitable for on-site detection compared to pre-trained models. Moreover, existing models have already demonstrated excellent performance in spectrum prediction tasks. Therefore, determining whether pre-trained models offer superior performance in this specific application scenario poses a challenge.

3. Regarding the issue of data leakage, the authors' response suggests that subtle structural differences in molecules can have a significant impact on MS2 outcomes. While this is an important consideration, it may be useful to provide further evidence or analysis to support this claim, especially since significant differences between highly similar molecules are often considered rare. As the MS will chunk the molecule into small fragments, many fragments might stay the same, and only one or two fragments that have small changes will only lead to small changes in one or two peaks.

Response:

We fully understand the reviewers' concerns regarding data leakage. When constructing our training dataset, we retained some highly similar molecules, ensuring that this did not result in any data leakage. It's important to note that for deep learning models, the input primarily comprises the spectra of molecules, and structural similarity between molecules doesn't necessarily imply high similarity in their tandem spectra.

For example, consider benzoic acid and para-benzoic acid, which exhibit highly similar molecular structures but significantly differ in their spectra. Similarly, luteolin and kaempferol, although structurally related, display distinct fragmentation peaks in their spectra. Experimental evidence reveals that while luteolin features peaks at $m/z=153,161,199,213,223$, kaempferol showcases peaks at $m/z=153,165,213,241,258$, with only $m/z=153$ being identical. Even

minor structural disparities can lead to substantial variations in the tandem spectra of substances.

Typically, secondary fragmentation spectra of substances, after excluding miscellaneous peaks, consist of 3-5 fragment peaks. The discrepancies between these peaks underscore significant differences between spectra. Therefore, retaining similar molecules during training enables the model to better discern the intricate relationship between spectra and molecular structures.

4. Lastly, the benchmark results against other deep learning methods in the same tasks suggest that there may be room for improvement in demonstrating a clear performance advantage.

Response:

We deeply understand their concerns regarding the model's performance. While our model may not exhibit exceptional performance advantages in predicting high-quality resolution spectra compared to other deep models, its focus lies in predicting tandem spectra with low resolution and accuracy. This capability enables effective on-site mass spectrum recognition. Models like MSNovelist rely on predicting molecular fingerprints for structure prediction, which imposes high requirements on mass accuracy. Similarly, the MassGenie model is geared towards recognizing spectra with high resolution and accuracy. Therefore, we devised a recognition model tailored for low-resolution spectra, effectively meeting the needs of on-site mass spectrometry and expanding its application scope.